# Nutrient density of Bangladeshi foods and its application in planning diet for pregnant women

**Nazma Shaheen**⊙*, **Abira Nowar, Saiful Islam**⊙, **Md. Hafizul Islam**⊙, **Md. Ruhul Amin**

Institute of Nutrition and Food Science, University of Dhaka, Dhaka, Bangladesh

* nazmashaheen@du.ac.bd

**Data Availability Statement:** The data underlying the results presented in the study will has been uploaded as the Supplementary Information.

**Funding:** The author(s) received no specific funding for this work.

## Abstract

Nutrient profiling is a method that classifies foods based on their nutrient content and identifies foods that are high in micronutrients both across and within food groups. This study aimed to identify foods that are rich sources of the seven micronutrients (iron, zinc, calcium, thiamine, riboflavin, vitamin A, and vitamin $B_{12}$) of public health concern for the Bangladeshi population._ This study developed a metric termed "naturally nutrient-rich score 7 ($NNR_7$)" specifically for third-trimester pregnant women to identify nutrient-dense foods. Further, it computed the nutrient adequacy score (NAS) of the top $NNR_7$-scored foods for seven micronutrients to assess the extent (percent) to which foods can meet pregnant women's recommended dietary allowances (RDA). A linear programming technique was then used to construct a nutrient-adequate model diet for third-trimester pregnant women using the top ten $NNR_7$-scored foods. According to the $NNR_7$, food groups such as leafy vegetables, fish, meat, poultry and eggs, and vegetables are the richest sources of the problem micronutrients. Mutton liver (916.7%), soybean (39.3%), lamb liver (2160%) and duck liver (50.0%) were found to fulfill the highest percentage of the RDA of vitamin A, zinc, vitamin $B_{12}$, and iron, respectively. In the formulated nutrient-adequate diets for pregnant women, rice, potato, brown wheat flour, and soya oil were universal to all three diets and Bengal gram, orange, Ganges River sprat, and duck liver were the most common ones. The study findings highlight the need for the consumption of foods such as leafy vegetables, fish, meat, poultry, eggs, pulses and vegetables to increase the intake of problematic micronutrients. Planning a nutrient-adequate diet for pregnant women using linear programming can be an alternative approach to optimize and shape food choices to meet their nutritional requirements.

## Introduction

Despite the tremendous progress in reducing the burden of malnutrition in the last two decades, micronutrient deficiencies, especially of vitamin A, iron (Fe), folic acid, calcium (Ca), zinc (Zn), and iodine, remain significant nutritional problems in Bangladesh [1]. A recent report on nutrient density and affordability of habitual diets in Bangladesh identified seven problem micronutrients for the Bangladeshi population which are Ca, Fe, Zn, vitamin $B_1$

**Competing interests:** The authors have declared that no competing interests exist.

**Abbreviations:** NNR7, Naturally nutrient-rich score 7; NAS, Nutrient Adequacy Score; RDA, Recommended dietary allowances; FCTB, Food Composition Table of Bangladesh; FBDG, Food-based dietary guidelines of Bangladesh; LP, Linear programming; icmr-NIN, Indian Council of Medical Research-National Institute of Nutrition; EER, Estimated energy requirement; EAR, Estimated Average Requirement; PAL, Physical activity level; TUL, Tolerable Upper Limit; AMDR, Acceptable Macronutrient Distribution Range.

(thiamine), vitamin $B_2$ (riboflavin), vitamin A, and vitamin $B_{12}$ [2]. These nutrients were termed "problem micronutrients" as they had a risk of inadequacy greater than 80% or more. These seven nutrients were also regarded as micronutrients of public health concern because they have a higher risk of inadequacy across all age groups, particularly among pregnant and lactating women, where the risk of inadequacy ranges from 80% to 100% [2].

Women during their pregnancies are more vulnerable to micronutrient deficiencies since they require more nutrients to meet the extra demands of gestation. Evidence suggests that maternal micronutrient deficiencies can impact the newborn from the beginning of their life by retarding fetal development to making them suffer from adverse health effects in later life [3]. Multiple studies have reported the consequences of maternal micronutrient deficiency such as deficiency of vitamin $B_{12}$ causing neural tube defect and poor fetal growth, zinc increasing the risk of preterm birth, and iron increasing the risk of low birth weight [3–6]. Supplementation of multiple micronutrients during this time has contributed substantially to reducing the risk of preterm delivery, being born with low birth weight, and being small for gestational age [3, 6–8]. However, better maternal diet quality characterized by a high intake of protein with essential micronutrient-dense vegetables, fruits, whole grains, and dairy products is considered to be one of the ideal vehicles for ensuring an adequate supply of micronutrients during pregnancy [9–13].

To tackle and control the burden of micronutrient malnutrition, the government of Bangladesh has undertaken community-based nutrition programs including community-based homestead gardening, the rearing of small livestock, and aquaculture [14]. The goals of these food-based initiatives are to boost the production of and availability of micronutrient-dense foods as well as marketing and education efforts to improve the consumption of such foods. To accomplish this, people need to be aware of the science-based tools and knowledge updates on nutrient-rich foods to make an informed choice of foods to fill their dietary nutrient gaps, especially micronutrients.

Among the different methods used to identify nutrient-rich foods, the naturally nutrient-rich (NNR) score stands out as a scientific scoring method for ranking foods based on their nutrient density [15]. This score can be used to assign nutrient density values to foods within and across food groups. The concept of NNR as a measure of nutrient density of foods was formed by Drewnowski Adam and the initial version of the index consisted of 14 nutrients. Later, the author added fiber and pantothenic acid and formulated the latest version of the index "Naturally Nutrient Rich score 16" [15]. Another method of assessing nutrient density is the calculation of the nutrient adequacy score (NAS), which evaluates a food's ability to meet the recommended dietary allowance (RDA) of selected nutrients in percentage [16]. Both NNR and NAS are science-based metrics that would help analyze diet quality and enable policy-makers, programmers, and consumers to identify nutrient-rich foods across different food groups. However, nutrient density analysis can only indicate whether a country's food system potentially supplies the foods with nutrients that are frequently lacking in the diets of vulnerable groups. It does not provide information on the appropriate combination of these available nutrient-rich foods required to quantitatively meet the demands for nutrients by individuals of specific vulnerable populations (young children, adolescents, pregnant and lactating women, etc.). Linear programming (LP) or careful meal planning may be effective in this situation [17].

LP is a mathematical approach with the potential to determine the appropriate selection of foods to achieve the nutrient and food intake recommendations. This technique has previously been applied in several instances, including nutritional quality improvement of food aid [18], optimization of cultural acceptability of diets [19, 20], and cost-minimization of healthy food baskets [21]. In Bangladesh, LP has previously been used, for example, to develop complementary feeding recommendations for children, and it was reported that a combination of local

foods could improve the nutrient density of diets but not entirely meet the children's nutrient requirements [22]. To our knowledge, however, in Bangladesh LP does not appear to be used to assess whether local foods combination can achieve the required nutrient density in the diets of vulnerable groups, such as pregnant women.

With this background, this study—being the first one of its kind to analyze the nutrient density of Bangladeshi foods—had three objectives. First, we aimed to estimate the nutrient density of the foods consumed by the Bangladeshi population and prioritize them based on their richness in the seven micronutrients frequently lacking in the diet of the Bangladeshi population. to accomplish this, we formulated a metric for third-trimester pregnant women named "Naturally Nutrient Rich 7 (NNR$_7$)" to identify foods that can meet their nutrient and energy demands. The reason for choosing pregnant women is that they are one of the most nutritionally vulnerable groups and that the inadequacy of the seven problem micronutrients was found to be the highest among them [2]. Second, we calculated nutrient adequacy scores (NAS) of the top ten NNR$_7$ scored foods from each food group to examine how well they could meet the RDA of the seven problem micronutrients. Finally, we formulated a model diet using LP that meets the nutrient and food intake recommendations for pregnant women as an application.

## Materials and methods

A nutrient density scoring matrix for a third trimester moderately active pregnant woman using the RDA of the seven problem micronutrients (calcium, iron, zinc, thiamine, riboflavin, vitamin A, and vitamin B$_{12}$) called "naturally nutrient-rich score 7 (NNR$_7$)" was formulated. The food composition data for each food was taken from the Food Composition Table of Bangladesh (FCTB) (S1 Appendix) [23].

In this study, all the foods of FCTB were divided into eight groups according to the food-based dietary guidelines of Bangladesh (FBDG), which are "cereals, starchy roots, and tubers", "pulses and legumes", "leafy vegetables", "non-leafy vegetables", "fish", "meat, poultry, and eggs", and "milk". Though the FBDG subsumes leafy and non-leafy vegetables in a single group, we separated them into different groups to identify naturally nutrient-rich foods from both groups individually. Similarly, we divided the "meat, fish, poultry, and eggs" group of the FBDG into two distinct groups ("fish" and "meat, poultry, and eggs") to identify the maximum numbers of naturally nutrient-rich foods from different food groups [24].

NAS was calculated for the top ten NNR$_7$ scored foods from each food group. These top ten NNR$_7$ scored foods from eight food groups along with one additional food (soya oil to appreciate cultural acceptability) were used as the food list for formulating a nutrient adequate diet for third-trimester pregnant women.

### Naturally Nutrient Rich Score (NNR$_7$)

NNR score is a nutrient density scoring matrix based on the daily values of the nutrients. The concept of NNR is basically a nutrients-to-calories ratio. Drewnowski formulated NNR score (i.e., NNR$_{14}$ and NNR$_{16}$) as average of percent daily values (DVs) for 14 and 16 nutrients, respectively [15]. In our calculation of NNR$_7$, we followed the same concept as Drewnowski [15]; However, we took seven nutrients as identified micronutrients of public health concern in Bangladesh. We also deviated by using RDA of the seven problem micronutrients instead of using their DVs. We could not use DVs as there is no standard DVs for nutrients for Bangladeshi population.

We defined NNR$_7$ as the mean of percent of RDAs for seven micronutrients in a 2480 kcal diet. As we aimed to identify naturally nutrient rich score foods rich in seven problem

**Table 1. List of seven nutrients and their recommended dietary allowance (RDA) of pregnant women used for calculating NNR$_7$ and NAS taken from ICMR-NIN.**

| Nutrients | Recommended Dietary Allowance (RDA) |
|---|---|
| Vitamin A (mcg RAE) | 900 |
| Thiamine (mg) | 2.0 |
| Riboflavin (mg) | 2.7 |
| Calcium (mg) | 1000 |
| Iron (mg) | 27 |
| Zinc (mg) | 14.5 |
| Vitamin B$_{12}$ (mcg) | 2.45 |

*Note*. RAE, Retinol activity equivalents; RDA = The recommended dietary allowance (RDA) is the average daily dietary intake level that suffices to meet the nutrient requirements of nearly all (97–98%) healthy persons of a specific sex, age, life stage, or physiological condition (such as pregnancy or lactation).

micronutrients for the third trimester pregnant women, we used their daily energy requirement (i.e., 2480 Kcal).

Firstly, we calculated the amount of a single food that would have to be consumed to get 2480 Kcal (i.e., the energy requirement of the reference population). We call this amount "2480-Kcal food."

Secondly, we calculated the contents of each of the problem nutrients in "2480-Kcal food" and expressed those as a percentage of their respective RDAs. We call each of those amounts "%RDA of a nutrient in 2480-Kcal food." If the "%RDA of a nutrient in 2480-Kcal food" was such that it exceeded 2000% of RDA for that nutrient, we truncated the percentage contribution at 2000% RDA so that any one nutrient does not unduly influence the final score [15]. The RDA values were taken from the Nutrient Requirements for Indians published by the Indian Council of Medical Research-National Institute of Nutrition (ICMR-NIN) [25] (Table 1).

Thirdly, we took the arithmetic average of %RDA of each of the seven problem nutrients in 2480-Kcal food by following Eq 1. We call this mean value the "Naturally Nutrient Rich Score 7" or "NNR$_7$."

$$NNR_7 = \frac{\sum_{n=1}^{n=7} \%RDA_{2480\ Kcal}}{7} \tag{1}$$

## Nutrient Adequacy Score (NAS)

NAS was calculated for examining to which extent (%) 100 g of food can fulfill the RDAs for each of the seven problem micronutrients. The following formula was used to calculate the NAS of foods.

$$NAS_i(\%) = \frac{Amount\ of\ nutrient_i\ in\ 100\ gm\ of\ food}{RDA\ of\ nutrient_i} \tag{2}$$

where i represents any of the seven problem micronutrients (Iron, Zinc, Calcium, Vitamin A, Thiamine, Riboflavin, and Vitamin B$_{12}$).

NAS was calculated for the eighty foods (top 10 NNR$_7$ scored from each food group) and the nutrient composition of foods was used from the FCTB. Similar to NNR$_7$, the RDA values were also used from ICMR-NIN for calculating NAS (Table 1).

## Linear programming to formulate an adequate diet for pregnant women

Linear programming (LP) maximizes, minimizes, or finds a specific value for an objective function that is defined by a set of decision variables and restricted by one or more linear constraints. In the present study, the goal of LP was to find a combination of foods (i.e., the decision variables) that meets the nutrients requirements (i.e., the linear constraints) while keeping the total dietary energy content (i.e., the objective function) equal to the estimated energy requirement (EER) for pregnant women. The resulting combination of foods was considered as the nutritionally adequate diet represented by a set of food weights (in grams). As defined mathematically below, the linear function set to be equal to the EER for pregnant women is the energy content of a nutritionally adequate diet for pregnant women.

$$\text{Energy content of diet} = W_1E_1 + W_2E_2 + W_3E_3 + \cdots + W_nE_n \tag{3}$$

Where $W_1$ to $W_n$ and $E_1$ to $E_n$ represent the weights and energy contents per unit weights of foods from 1 to n, respectively.

It is well-known that energy recommendations vary with physical activity level (PAL); We, therefore, assessed whether locally available foods could meet the required nutrient density of the diets across three energy levels for pregnant women engaged in *sedentary* (e.g., desk work mostly in sitting), *moderate* (standing, walking, cycling, and carrying light weights) and *heavy* (e.g., agriculture and mine workers carrying loads and pulling carts) activity. The EER values for pregnant women with *light*, *moderate* and *heavy* PAL were 2010, 2480, and 3070 Kcal, respectively, calculated by the ICMR-NIN Expert Group on Nutrient requirement for Indians in 2020 [25].

The linear function defined above was subject to three sets of constraints as shown in Table 2: micronutrients constraints, macronutrients constraints, and food (groups) constraints. The Estimated Average Requirement (EAR) of all the seven micronutrients (Ca, Fe, Zn, vitamin A, thiamine, riboflavin, and vitamin $B_{12}$) was set up as 'greater than or equal' inequality. However, for micronutrients with a maximum tolerable limit (i.e., Ca, Fe, Zn, and vitamin A), a Tolerable Upper Limit (TUL) was set up as 'less than or equal' inequality. Mathematically,

$$W_1m_{i1} + W_2m_{i2} + W_3m_{i3} + \ldots + W_nm_{in} \geq EAR_i \tag{4}$$

where W1, W2, . . . Wn are the amounts of foods 1 to n; $m_{i1}, m_{i2}, \ldots m_{in}$ are the concentrations of micronutrient i in foods 1 to n; and i = 1, 2, . . .,7; and

$$W_1m_{j1} + W_2m_{j2} + W_3m_{j3} + \ldots + W_nm_{jn} \leq TUL_j \tag{5}$$

where W1, W2, . . .Wn are the amounts of foods 1 to n; $m_{j1}, m_{j2}, \ldots m_{jn}$ are the concentrations of micronutrient j in foods 1 to n; and j = 1, 2, . . .,4.

For macronutrients, the inequality constraints were set as follows using the Acceptable Macronutrient Distribution Range (AMDR) of carbohydrates, proteins, and fats to limit the proportion of EER coming from different macronutrients:

$$L_k \leq W_1M_{k1} + W_2M_{k2} + W_3M_{k3} + \ldots + W_nM_{kn} \leq U_k$$

where W1, W2, . . .Wn are the amounts of foods 1 to n; $M_{k1}, M_{k2}, \ldots M_{kn}$ are the concentrations of micronutrient k in foods 1 to n; k = 1, 2, 3; $L_k$ and $U_k$ are the lower and upper end of the AMDR of macronutrient k.

Similar to EER, the values for EAR, TUL, and AMDR were those recommended by the ICMR-NIN Expert Group on Nutrient requirement for Indians in 2020 [25]. Constraints for

**Table 2. Inequality constraints used to construct a nutritionally adequate diet for third trimester pregnant women in Bangladesh.**

| Constraints | | Minimum | Maximum |
|---|---|---|---|
| Micronutrients | Iron | 21 mg | 45 mg |
| | Calcium | 800 mg | 2500 mg |
| | Zinc | 12 mg | 40 mg |
| | Thiamine | 1.6 mg | - |
| | Riboflavin | 2.3 mg | - |
| | Vitamin B$_{12}$ | 2.2 mcg | - |
| | Vitamin A | 406 mcg | 3000 mcg |
| Macronutrients | Carbohydrate | 45% of EER | 65% of EER |
| | Protein | 53.6 g | 15% of EER |
| | Fat | 15% of EER | 15% of EER |
| Food groups | Cereals | 270 g | 450 g |
| | Leafy vegetables | 150 g | 300 g |
| | Non-leafy vegetables | 150 g | 300 g |
| | Fruits | 100 g | 200 g |
| | Pulses | 30 g | 60 g |
| | Meat, poultry, and egg | 100 g | 250 g |
| | Fish | 50 g | 100 g |
| | Milk and milk products | 1 serving (or 100 Kcal equivalent) | 3 servings (or 300 Kcal equivalent) |
| Foods | Rice | 33% of EER | - |
| | Potato | 50 g | 200 g |
| | Soyabean oil | 15 g | 30 g |
| | Wheat | 100 g | 200 g |

*Note.* EER = Estimated energy requirement

food groups, however, were set to reflect the amounts of foods from different food groups as recommended in the National Dietary Guidelines in Bangladesh [24].

Mathematically,

$$\text{Min}_g \leq \sum_{l=1}^{10} w_{gl} \geq \text{Max}_g$$

Where Min$_g$ and Max$_g$ are the minimum and maximum amounts of foods respectively from the food group g; l = 1, 2, 3, . . .10 (i.e., top 10 foods based on NNR$_7$ score within each food group); and g = 1, 2, 3, . . .8 (i.e., eight food groups). In addition to the eighty foods (resulting from 10 foods within each of the eight food groups), we included soya oil to achieve a diet that were more practical for recommending to pregnant women in Bangladesh. Based on the typical food consumption behavior as reflected in the latest dietary intake survey in Bangladesh, we included four additional food constraints for rice, potato, wheat and soya oil so that the diet contains at least one-third of EER from rice, 100–200 grams of wheat, 50–200 grams of potato, and 15–30 grams of soya oil. On the other hand, the amount of sweetened condensed milk within the food group milk and milk products was set to zero giving an effective final list of decision variables of 81 foods (S2 Appendix). We used the Simplex algorithm available in the in-built Solver optimizer of Microsoft Excel for Windows to formulate the desired diet through LP. In the results of the LP, we did not report foods which resulted in amount less than 5 g.

**Table 3. Median NNR$_7$ scores of eight food groups with their minimum and maximum values.**

| Food groups | NNR$_7$ score<br>Median (minimum, maximum) |
|---|---|
| Cereals, starchy roots, and tubers (n = 66) | 42.8 (4.8, 334.9) |
| Pulses and legumes (n = 18) | 81.5 (61.1, 134.1) |
| Vegetables (n = 57) | 170.3 (36.1, 565.9) |
| Leafy vegetables (n = 39) | 647.1 (191.2, 1095.3) |
| Fruits (n = 46) | 76.9 (22.9, 483.4) |
| Fish (n = 79) | 221.3 (26.7, 678.7) |
| Meat, poultry, and egg (n = 26) | 208.6 (57.9, 986.1) |
| Milk (n = 13) | 149.2 (66.1, 555.9) |

*Note.* NNR$_7$ = Naturally Nutrient Rich Score 7

## Results

According to NNR$_7$, leafy vegetables (647.1) was the most naturally nutrient-rich food group followed by fish (221.3), meat, poultry, and egg (208.6), and vegetables (170.3). Milk had an NNR$_7$ score of 149.2 and the lowest median score belonged to the cereals, starchy roots, and tubers food group (Table 3).

Table 4 presents the NAS of the top five foods from each food group for seven problem nutrients. According to the table, duck liver can fulfill the highest percentage (50%) of the RDA of iron, whereas soybean can fulfill the highest percentage (39.3%) of the RDA for zinc.

**Table 4. List of top five foods based on Nutrient Adequacy Score (NAS) for pregnant women based on the Recommended Daily Allowance (RDA) from icmr-NIN.**

| Food groups | Nutrients | Food name (% fulfillment of RDA of pregnant women from 100 g food) |
|---|---|---|
| Cereals, starchy roots, and tubers | Iron | Pearl millet (29.6%), barley (20.2%), proso millet (18.5%), wheat flour (18.1%), wheat (18.1%) |
| | Zinc | Pearl millet (21.4%), wheat flour (20.8%), wheat (19.2%), barley (19.1%), proso millet (15.9%) |
| | Calcium | Bread (white, toasting) (11.9%), wheat flour (5.2%), elephant foot (5.0%), giant taro (4.6%), pear millet (4.2%) |
| | Vitamin A | Sweet potato (orange flesh) (79.9%), pear millet (0.3%), potato (diamond) (0.3%), barley (0.1%), wheat flour (0.1%), wheat (0.1%) |
| | Thiamine | Wheat flour (24.5%), wheat (24.5%), barley (23.5%), proso millet (21.1%), pear millet (16.5%) |
| | Riboflavin | Bread (white, toasting) (12.6%), proso millet (10.7%), barley (7.4%), wheat flour (6.3%), pear millet (5.9%) |
| Pulses and legumes | Iron | Soybean (42.0%), Bengal gram (dehulled) (33.0%), bengal gram (whole) (32.7%), green gram (whole) (29.0%), green gram (split) (24.1%), cowpea (seed) (24.1%) |
| | Zinc | Soybean (39.3%), lentil (26.8%), pea (24.1%), grass pea (split) (23.3%), cowpea (seed) (23.3%) |
| | Calcium | Soybean (24.1%), Bengal gram (whole) (20.3%), black gram (dehulled) (18.4%), green gram (whole) (13.7%), pea (7.5%) |
| | Vitamin A | Bengal gram (dehulled) (1.1%), grass pea (split) (0.6%), green gram (split) (0.4%), Bengal gram (whole) (0.4%), green gram (whole) (0.4%) |
| | Thiamine | Lentil (38.5%), soybean (36.5%), green gram (whole) (23.5%), pea (23.5%), black gram (dehulled) (21.0%) |
| | Riboflavin | Soybean (18.5%), green gram (whole) (14.4%), Bengal gram (whole) (9.8%), black gram (dehulled) (9.6%), Bengal gram (dehulled) (7.8%) |

*(Continued)*

**Table 4.** (Continued)

| Food groups | Nutrients | Food name (% fulfillment of RDA of pregnant women from 100 g food) |
|---|---|---|
| Non-leafy vegetables | Iron | Amaranth stem (6.7%), pointed gourd (6.3%), onion (3.3%), bean (3.3%), okra (3.3%) |
| | Zinc | Amaranth stem (3.6%), onion (2.8%), pointed gourd (2.8%), radish (2.6%), bean (2.6%) |
| | Calcium | Amaranth stem (11.4%), okra (9.3%), bean (7.0%), pumpkin (5.2%), ash gourd (3.0%) |
| | Vitamin A | Carrot (49.4%), pumpkin (41.0%), amaranth stem (2.8%), bean (2.1%), okra (2.1%) |
| | Thiamine | Radish (21.5%), pointed gourd (8.5%), cucumber (8.0%), bean (4.0%), pumpkin (3.5%) |
| | Riboflavin | Amaranth stem (6.7%), okra (5.9%), onion (5.2%), bean (3.3%), carrot (3.0%) |
| Leafy vegetables | Iron | Spiney amaranth leaves (53.3%), jute leaves (35.9%), green amaranth leaves (31.1%), Bengal dayflower leaves (26.3%), red amaranth leaves (22.2%) |
| | Zinc | Jute leaves (10.1%), bathua leaves (6.8%), green amaranth leaves (6.7%), red amaranth leaves (6.6%), mustard leaves (4.7%) |
| | Calcium | Spiney amaranth leaves (64.0%), fenugreek leaves (39.5%), red amaranth leaves (25.6%), beet greens leaves (24.9%), bathua leaves (21.1%) |
| | Vitamin A | Spiney amaranth leaves (100.9%), red amaranth leaves (88.1%), fenugreek leaves (84.3%), green amaranth leaves (82.6%), beet greens leaves (51.5%) |
| | Thiamine | Beet greens leaves (9.0%), fenugreek leaves (6.0%), jute leaves (5.0%), mustard leaves (4.0%), bathua leaves (3.0%) |
| | Riboflavin | Jute leaves (20.2%), bathua leaves (18.9%), beet greens leaves (14.4%), Indian spinach (13.3%), fenugreek leaves (11.5%) |
| Fruits | Iron | Hog plum (10.4%), fig (4.1%), jambolan (3.0%), monkey-jack (2.9%), elephant apple (2.6%) |
| | Zinc | Monkey-jack (11.6%), mango (6.0%), elephant apple (2.6%), jambolan (1.4%), papaya (1.2%) |
| | Calcium | Fig (8.0%), elephant apple (7.4%), monkey-jack (5.7%), hog plum (5.7%), papaya (2.9%) |
| | Vitamin A | Monkey-jack (34.4%), mango (32.5%), melon (11.6%), jambolan (10.3%), papaya (6.7%) |
| | Thiamine | Elephant apple (40%), hog plum (14%), melon (5.5%), muskmelon (5.5%), jambolan (4.5%) |
| | Riboflavin | Monkey-jack (8.5%), melon (3.0%), muskmelon (3.0%), fig (1.9%), hog plum (1.5%) |
| Fish | Iron | Minnow largescale razorbelly (20.0%), Indian river shad (17.9%), mola carplet (14.0%), day's mystus (10.4%), pool barb (9.6%) |
| | Zinc | Mola carplet (22.0%), minnow finescale razorbelly (21.4%), ganges river sprat (21.4%), minnow largescale razorbelly (21.4%), pool barb (20.7%) |
| | Calcium | Indian river shad (106.0%), pool barb (96.7%), mola carplet (76.7%), rohu (65%), day's mystus (62.7%) |
| | Vitamin A | Mola carplet (297.8%), spotted snakehead (21.2%), pool barb (6.6%), day's mystus (4.8%), ganges river sprat (4.2%) |
| | Thiamine | Bronze featherback (6.0%), minnow largescale razorbelly (4.7%), rohu (2.5%), spotted snakehead (2.0%), ganges river sprat (1.5%) |
| | Riboflavin | Minnow largescale razorbelly (4.0%), bronze featherback (3.0%), rohu (3.6%), spotted snakehead (1.9%), ganges river sprat (1.9%) |
| | Vitamin $B_{12}$ | Mola carplet (325.7%), Indian river shad (285.3%), minnow largescale razorbelly (230.2%), minnow finescale razorbelly (230.2%), roh (204.1%) |

(*Continued*)

**Table 4.** (Continued)

| Food groups | Nutrients | Food name (% fulfillment of RDA of pregnant women from 100 g food) |
|---|---|---|
| Meat, poultry, and egg | Iron | Duck liver (50.0%), chicken liver (33.3%), mutton liver (23.3%), hen egg yolk (17.7%), beef liver (13.0%) |
| | Zinc | Beef (lean, minced) (35.1%), mutton liver (27.6%), beef liver (25.6%), hen egg yolk (23.0%), beef meat (23.0%) |
| | Calcium | Hen egg yolk (12.0%), duck egg (whole) (6.5%), duck liver (3.3%), hen egg (whole) (2.9%), chicken liver (1.8%) |
| | Vitamin A | Mutton liver (916.7%), beef liver (552.0%), chicken liver (366.2%), duck liver (93.8%), hen egg yolk (55.1%) |
| | Thiamine | Mutton liver (28.0%), duck meat (28.0%), duck liver (17.5%), chicken liver (15.3%), hen egg yolk (11.6%) |
| | Riboflavin | Beef liver (102.0%), chicken liver (65.9%), mutton liver (63%), duck liver (41.1%), hen egg yolk (18.1%) |
| | Vitamin $B_{12}$ | Mutton liver (2204.1%), chicken liver (1428.6%), duck egg (whole) (220.4%), hen egg yolk (136.3%), beef meat (134.7%) |
| Milk | Iron | Skimmed powdered milk (4.1%), whole powdered milk (2.4%), cottage cheese (1.1%), condensed milk (0.8%), goat milk (0.7%) |
| | Zinc | Skimmed powdered milk (32.4%), whole powdered milk (25.6%), cottage cheese (24.5%), condensed milk (6.7%), curd (3.1%), cow's milk (skimmed) (3.1%), cow's whole milk (whole fat) (3.1%) |
| | Calcium | Skimmed powdered milk (137%), whole powdered milk (95.9%), cottage cheese (79.0%), condensed milk (28.7%), buffalo milk (20.6%) |
| | Vitamin A | Whole powdered milk (26.4%), cottage cheese (22.8%), condensed milk (10.6%), buffalo milk (5.2%), curd (3.6%) |
| | Thiamine | Skimmed powdered milk (22.5%), whole powdered milk (15.6%), condensed milk (4.5%), goat milk (4.5%), cow's milk (skimmed) (3.1%), cow's milk (whole fat) (3.1%) |
| | Riboflavin | Skimmed powdered milk (60.7%), whole powdered milk (50.6%), cottage cheese (17.3%), condensed milk (16.3%), goat milk (16.3%) |
| | Vitamin $B_{12}$ | Whole powdered milk (132.6%), cow's milk (skimmed) (20.0%), buttermilk (13.9%), cow's milk (whole fat) (10.2%), curd (10.2%) |

For calcium, Indian river shad showed a NAS of 106%, which means that 100 g of it can fulfill 106% of the daily allowance of calcium. Similarly, it can be seen from the table that foods such as mutton liver (916.7%), elephant apple (40%), beef liver (102%) fulfill the highest percentage of the RDA of vitamin A, thiamine, and riboflavin, respectively. As vitamin $B_{12}$ can only be found from animal source foods, the highest percentage of the RDA of vitamin $B_{12}$ can be fulfilled by mutton liver (2204.1%) followed by chicken liver (1428.6%) and mola carplet (an indigenous bony fish of Bangladesh) (325.7%).

The nutrient adequate diet formulated for third-trimester pregnant women using linear programming are presented in Table 5. Rice, brown wheat flour, soya oil, and potato were universal to all three diets for different activity levels. The diets contained foods from eight food groups, and the most common foods were Bengal gram, orange, Ganges River sprat, and duck liver. These foods varied in amounts with the different activity levels. For example, a sedentarily active woman would need to eat 267 g of rice, whereas moderately and heavily active would have to eat 300 g and 513 g, respectively.

## Discussion

To identify foods that are rich sources of the seven problem micronutrients of Bangladesh, this study constructed a metric, "naturally nutrient-rich score 7 ($NNR_7$)" specifically for third-

**Table 5. Nutrient adequate diet for pregnant women of different activity levels through linear programming.**

| Sedentarily active (2010 kcal) | | Moderately active (2480 kcal) | | Heavily active (3070 kcal) | |
|---|---|---|---|---|---|
| Food items | Weight (g) | Food items | Weight (g) | Food items | Weight (g) |
| Rice | 267 | Rice | 300 | Rice | 513 |
| Brown wheat flour | 100 | Brown wheat flour | 100 | Brown wheat flour | 100 |
| Potato | 50 | Potato | 50 | Potato | 50 |
| Bengal gram | 30 | Grass pea | 58 | Bengal gram | 30 |
| Pointed gourd | 150 | Carrot | 150 | Amaranth stem | 150 |
| Jute leaves | 150 | Indian spinach | 150 | Bathua leaves | 150 |
| Orange | 100 | Monkey-jack | 200 | Orange | 100 |
| Ganges river sprat | 37 | Bronze featherback | 50 | Ganges river sprat | 50 |
| Indian river shad | 12 | Duck liver | 10 | Minced beef | 34 |
| Beef liver | 46 | Chicken egg | 54 | Chicken egg yolk | 110 |
| Duck liver | 22 | Chicken egg yolk | 32 | Cottage cheese | 18 |
| Other bird's meat (Quail) | 30 | Curd | 67 | Soya oil | 22 |
| Buffalo milk | 22 | Soya oil | 30 | - | - |
| Skimmed powdered milk | 11 | - | - | - | - |
| Soya oil | 30 | - | - | - | - |

*Note.* Sedentarily active = desk work mostly in sitting

Moderately active = standing, walking, cycling, and carrying light weights

Heavily active = agriculture and mine workers carrying loads and pulling carts

trimester pregnant women in Bangladesh. It also assessed the NAS of the top ten $NNR_7$-scored foods across the food groups (n = 80) to assess what percent of the RDAs of the micronutrients can be fulfilled by consuming 100 g of food. Using the top $NNR_7$-scored foods, the study formulated a nutrient-adequate diet for third-trimester pregnant women of different activity levels using linear programming.

Pregnancy is a period when women have excess demand for nutrients to meet the growth and development of the fetus. Nutrient deficiency at this stage can lead to adverse birth outcomes and increases the risk of developing cardiovascular, pulmonary, and metabolic diseases in the long run [10]. Therefore, this study developed a metric particularly for pregnant women and identified food groups that are the richest sources of the seven problem micronutrients. The study identified food groups such as leafy vegetables, fish, meat, poultry and eggs, and pulses and vegetables as the richest sources of micronutrients with respect to the seven problem micronutrients. Similar to the study results, a study conducted by Connell et al. (2012) reported vegetables as the richest sources of micronutrients followed by fruits [26]. Thus, the study results denote that to reduce the prevalence and 'risk of inadequacy' of these nutrients among pregnant women, we need to increase the consumption of vegetables and protein-rich foods in our habitual diets. However, the actual scenario is not satisfactory. According to the latest household income and expenditure survey, 2022 the average per capita per day consumption of fish, meat, and egg of Bangladeshi population is only 67.8 g, 40.0 g, and 12.7 g, respectively. The habitual diet is still dominated by rice which contains one third portion of the total diet (328.9 g) [27]. With this scenario, this method of nutrient density scoring can be applied in designing meals, menus, and diets for populations belonging to different age groups. The method of nutrient profiling also shows how the notion of nutrient density goes into total diet quality and the economic aspect of food selection factors [28]. Despite all the advantages and following the scientific process to score foods, nutrient profiling method has a limitation.

The method has no standard reference or benchmark of scores to classify foods into categories like high, medium, and low nutrient-dense ones.

In addition to identifying nutrient-rich food groups for pregnant women, the study also assessed the nutrient adequacy of the top ten $NNR_7$ scored foods from each food group. These scores can help consumers to know what foods can meet the maximum percentage of their daily requirements for the seven nutrients. For example, in this study among foods from all the food groups, mutton liver could fulfill 916.7% of the RDA of vitamin A and soybean can fulfill 39.3% of the zinc. Animal-based foods like fish, eggs, meat, and animal liver were found to be the major sources of the problem micronutrients, especially for vitamin $B_{12}$, which is found in animal source foods only. Though NAS is a way of evaluating the nutritional adequacy of foods, this method is not as comprehensive as nutrient density scoring because it considers single nutrient requirements [29]. However, it can be used as an important tool to identify foods that can meet the requirements of a specific nutrient and fill the nutrient gap of the population through diet. As the article provided NAS of top ten foods from each food group, this list can be used as an exchange list across and within the food groups. Food exchange list is an extensively used dietary tool that helps individuals manage their dietary behavior and modify diets for any disease or physiological condition. This implies if a pregnant woman has iron deficiency, she may choose foods from meat, poultry, and egg as 100g of these foods fulfill the highest percentage of their iron requirement. In addition, she would have the flexibility to choose foods alternatively among duck liver, chicken liver, mutton liver, egg yolk, and beef liver within the same food group.

In our analysis, we formulated nutrient adequate diets by linear programming using the top ten $NNR_7$ scored foods from each food group. Linear programming is a method that helps to formulate diets meeting all the given nutritional constraints by minimizing or maximizing a decision variable [19]. Several studies have used linear programming to formulate country-specific balanced diets using locally available and culturally accepted foods [20, 30, 31]. A study conducted in Myanmar used the linear programming approach and reported anchovy, chicken liver, beans, and lentil as rich food sources to fill the gap of problem nutrients [32]. The study chose 12–23 months old children as their reference population to provide diets for improving their complementary feeding. Similarly, our study chose third-trimester pregnant women as the reference population given that women often enter pregnancy malnourished, and the demands of gestation can exacerbate micronutrient deficiencies with health consequences to the fetus [3, 33]. Moreover, according to the recent report by Shaheen et al. (2021), this reference population has the highest risk of inadequacy for the seven problem micronutrients [34]. Diets were formulated for three different activity levels as there is a difference in energy requirements among sedentary, moderate, and heavily active pregnant women. Brown rice was universal to all diets, and we included potato and soya oil in the diets as these two items are part of the habitual diets of Bangladesh.

The concept of nutrient profiling and ranking the foods according to the nutrient density score can be a useful method for the Bangladeshi population to choose nutrient-rich foods that fulfill their requirements [35]. Different nutrition programs with a focus to improve maternal health may utilize the nutrient density and nutrient adequacy scores to design diets for pregnant women. Moreover, these scores can also be used to inform pregnant women about the nutrient rich foods and assist them to make educated food choices. The diet formulated by linear programming can be used as an example for diet or menu planning during pregnancy in hospitals, diet counselling centers, and research organizations. Future studies may use linear programming to plan minimum cost diets that meet all nutritional requirements of individuals across different life cycle stages. Researchers may also conduct studies to validate the formulated model diets. Using the concept of nutrient scoring, studies may analyze $NNR_7$ for other

vulnerable groups such as under five children and adolescents and identify foods are the rich sources of the problem micronutrients. Further studies may also include other ethnic and local indigenous foods in their analysis which may not be present in the food composition table.

The current study has the advantage of being the first to develop a nutrient density scoring metric for third-trimester pregnant women to score foods based on their richness in the identified seven problem micronutrients of Bangladesh. The major strength of the study is that it constructed a scoring metric focusing on a specific population. The merit of the study is that it not only provides information on foods that can be used to combat deficiency of the seven micronutrients among pregnant women but also it shows which foods can fulfill their requirement at the maximum. The study also formulated a nutrient adequate diet for third-trimester pregnant women which can be used as a sample diet for the reference population. The limitation of the study is that it may have missed out some ethnic and locally available foods as it only analyzed foods that are available in the FCTB for Bangladesh.

## Conclusions

The study constructed a metric called $NNR_7$ to identify foods which are the richest sources of the seven problem micronutrients for a third-trimester pregnant woman. The findings demonstrated that the consumption of foods such as leafy vegetables, fish, meat, poultry, and eggs, as well as pulses and vegetables, must be increased in Bangladeshi habitual diets to secure the supply of problematic micronutrients. In addition, the study used pregnant women as the reference group and nutrient adequacy results showed that mutton liver, soybean, and Indian river shad could fulfill the highest percentage of daily allowances of vitamin A, zinc, and calcium. The use of linear programming in formulating diets can be used to design nutritionally adequate diets for populations of different age groups.

## Supporting information

**S1 Appendix. Food composition data of the foods for calculating $NNR_7$.**
(XLSX)

**S2 Appendix. List of foods for formulating a nutrient adequate diet for pregnant women.**
(DOCX)

## Author Contributions

**Conceptualization:** Nazma Shaheen, Md. Ruhul Amin.

**Data curation:** Abira Nowar, Md. Hafizul Islam.

**Formal analysis:** Abira Nowar, Saiful Islam, Md. Hafizul Islam.

**Investigation:** Nazma Shaheen.

**Methodology:** Nazma Shaheen, Saiful Islam, Md. Ruhul Amin.

**Project administration:** Nazma Shaheen.

**Software:** Saiful Islam.

**Supervision:** Nazma Shaheen, Md. Ruhul Amin.

**Visualization:** Nazma Shaheen.

**Writing – original draft:** Abira Nowar.

**Writing – review & editing:** Nazma Shaheen, Abira Nowar, Saiful Islam, Md. Hafizul Islam, Md. Ruhul Amin.

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
