## [Decision Letter · Decision Letter 0]

24 Jul 2023

PONE-D-23-13289Nutrient Density of Bangladeshi foods and its application in diet planning for pregnant womenPLOS ONE

Dear Dr. Shaheen,

Thank you for submitting your manuscript to PLOS ONE. After careful consideration, we feel that it has merit but does not fully meet PLOS ONE’s publication criteria as it currently stands. Therefore, we invite you to submit a revised version of the manuscript that addresses the points raised during the review process.

We look forward to receiving your revised manuscript.

Kind regards,

Dharmendra Kumar Meena

Academic Editor

PLOS ONE

Journal Requirements:

“We acknowledge the University of Dhaka for providing the support for APC”

Additional Editor Comments:

article still lacking of language editing in terms of phrasing, grammar and sentence framing.

Reviewers' comments:

Reviewer's Responses to Questions

**Comments to the Author**

1. Is the manuscript technically sound, and do the data support the conclusions?

Reviewer #1: Yes

Reviewer #2: Yes

Reviewer #3: Yes

Reviewer #4: Yes

2. Has the statistical analysis been performed appropriately and rigorously? 

Reviewer #1: Yes

Reviewer #2: I Don't Know

Reviewer #3: Yes

Reviewer #4: Yes

3. Have the authors made all data underlying the findings in their manuscript fully available?

Reviewer #1: Yes

Reviewer #2: Yes

Reviewer #3: Yes

Reviewer #4: Yes

4. Is the manuscript presented in an intelligible fashion and written in standard English?

Reviewer #1: Yes

Reviewer #2: Yes

Reviewer #3: Yes

Reviewer #4: Yes

5. Review Comments to the Author

Reviewer #1: The idea is though not new but unique in the context of Bangladesh needs. The model is explained well, enough for the readers from health and nutrition background. But if these two suggestions are added the material given in the article will become more understandable.

1. Was a software used for this intense linear programing? if yes please mention its name. If no, then supplementary material for all manual calculation may please be added.

2. The decision matrix of LP may also be added in supplementary material.

Thirdly; if the equations expressed at line # 143, 158, 174, 177, and 186 and 195 are labelled as equation no.1, 2, 3, etc., will be better presentation of the mathematical work.

Reviewer #2: The idea of NNR7 looks like a good new method; it is statistically easy to perform and will show the lack of the identified problem micronutrients. In the future, it will be applied to other demographic groups.

Reviewer #3: PLOS ONE

Manuscript Number: PONE-D-23-13289

Nutrient Density of Bangladeshi foods and its application in diet planning for pregnant women

Since the paper involves a large number of abbreviations for simplicity, it is recommended that a list of such abbreviations be included.

Nutrient Adequacy Score (NAS)

Naturally nutrient-rich score 7 (NNR7)

Recommended dietary allowances (RDA)

being small for gestational age (SGA)

Food Composition Table of Bangladesh (FCTB)

Food-based dietary guidelines of Bangladesh (FBDG)

Linear programming (LP)

Estimated energy requirement (EER)

Physical activity level (PAL)

Tolerable Upper Limit (TUL)

Acceptable Macronutrient Distribution Range (AMDR)

It is necessary that each equation be assigned a number.

Folic acid deficiency is typically regarded to be more common than calcium deficiency among pregnant women Inadequate folic acid intake during pregnancy is connected with a number of potential health problems. What are these risks? Why the authors were folic acid excluded from the study, even though the need for folic acid is important at all stages of pregnancy? As folic acid is no less important than calcium, which was mentioned in this study.

In line 26 NNR7,98, 108, 238, 246, 308, 318 NNR7, these terms new naturally nutrient-rich score, Naturally Nutrient Rich 7, new Nutrient-Rich Foods (NRF), and nutrient density scoring matrix are not new and you should specify with new metric for pregnant women in Bangladesh.

Actually, these terms had been used in various research studies and publications to assess the nutritional quality of different foods and dietary patterns. The United States Department of Agriculture (USDA) created a nutrient profiling technique known as the Nutrient-Rich Foods Index in order to evaluate the overall nutritional content of different foods. It assigns a ranking to each food item based on its nutrient density, considering both the good and negative aspects of the foods' nutrient composition when doing so.

In line 27, the phrase “third trimester pregnant women” should be added to the “specifically for pregnant women…”.

In line 62, remove the short letters of the being small for gestational age (SGA) as it didn’t repeat it within entire manuscript.

In line 100, the authors only targeted the later stages of pregnancy; they did not target the entire gestation period. Therefore, only referring to pregnant women in their third trimester would be the most accurate course of action to pursue.

In line 115, delete “and” of the and egg.

In line 117-120, the sentence is needs reference (such as FAO or FANTA,...etc).

In line 134 table 1, the references should be mention the recommended dietary allowance of pregnant women and which allowances were followed (icmr-NIN and/or others).

In line 146, are the authors followed a specific food group classification. The classification of food groups can vary depending on the system or organization you're should referring to! However, one common classification system is based on the United States Department of Agriculture (USDA) food groups, which are used for dietary guidelines and nutritional recommendations.

In line 163, the word light should be change with sedentary.

In line 172, the phrase ‘adverse effects on excess consumption’ is belong to the pregnant women take any supplements or medication or from the natural food? What grad of the adverse effect from natural food? I would suggest to remove these phrases or justified.

In line 223, in the table 4 authors should mention to the Indian icmr-NIN as they considered as a RDA reference

In line 224 table 4, food groups. Thiamine is a vitamin that can be degraded when exposed to high temperatures since it is heat-sensitive. However, the amount of thiamine that actually breaks down during baking might be somewhat variable. Such as... * Note: Thiamine is sensitive to heat; therefore, grains and baked goods are not considered the primary sources of thiamine.

Reviewer #4: Dear Editor,

I apologize for the delay in my review of the manuscript. Please see my comments below.

I read the manuscript with interest. The manuscript reports the nutrient profiling of local foods of Bangladesh to prioritize foods based on their nutrient density and plan sample diet to meet pregnant women's requirements.

Comment 1: The authors have claimed that the metric used to calculate the nutrient density of foods is completely new. NNR7 is not a completely new metric, it's an NNR version in which the nutrients selected are different from the original version. I suggest citing and explaining the original version of NNR (Drewnowski A. Concept of a nutritious food: Toward a nutrient density score. Am J Clin Nutr.368 2005;82(4):721–32) first (in the method section) and then explaining how and why this version (NNR7) was developed in detail.

Comment 2: Please clarify how NNR has been calculated step by step. The following is the original formula developed by Drewnowski (Drewnowski A. Concept of a nutritious food: toward a nutrient density score. Am J Clin Nutr 2005;82(4):721-32.)

NNR=Σ %DV2000 kcal /14

the formula presented here could not be correct. NNR is the mean of DV for selected nutrients, not RDA.

Comment 3. In this study, three nutrient-adequate diets have been formulated for third-trimester pregnant women. Although the method used to produce these diets is interesting, I think the formulated diets are not that applicable, even as an example. The foods and the amounts calculated by linear programming may not practically form a real daily menu. The alternative approach could be calculating the amount of serving size for each food group, then highlighting the food with the highest nutrient density in each food group to fulfill requirements for selected nutrients. If such results are achievable by linear programming, I recommend reporting them instead of what has been presented as a nutrient-adequate diet.

6. PLOS authors have the option to publish the peer review history of their article (what does this mean?). If published, this will include your full peer review and any attached files.

Reviewer #1: **Yes: **Muhammad Aasim

Reviewer #2: **Yes: **Ahmad M. Al Athamneh

Reviewer #3: **Yes: **Mustafa Alsharafani

Reviewer #4: No

---

## [Author Response · Author response to Decision Letter 0]

20 Sep 2023

Reviewer 1 

Comment 1: Was a software used for this intense linear programing? if yes please mention its name. If no, then supplementary material for all manual calculation may please be added.

Author’s response: Thank you for pointing out this omission. Although not explicitly mentioned in the manuscript, we used Microsoft Excel for Windows and its in-built Solver optimizer for performing linear programming. The revised manuscript now reads “We used the Simplex algorithm available in the in-built Solver optimizer of Microsoft Excel for Windows to formulate the desired diet through LP ” as can be seen in lines 223 and 224. 

Comment 2: The decision matrix of LP may also be added in supplementary material.

Author’s response: Thank you for your suggestion. The complete list of foods used as the input to the LP model is provided in the supplementary material (S2 Appendix). However, the results of decision variables (i.e., the amounts of foods from different food groups) are provided in Table 5. Please note that the constraints are also listed in Table 2.

Comment 3: Thirdly; if the equations expressed at line # 143, 158, 174, 177, and 186 and 195 are labelled as equation no.1, 2, 3, etc., will be better presentation of the mathematical work.

Author’s response: Thank you for your comment. We have inserted numbers to the equations in the revised manuscript as can be seen in line numbers 149, 160, 176, 192, and 195.

Reviewer 2

Comment: The idea of NNR7 looks like a good new method; it is statistically easy to perform and will show the lack of the identified problem micronutrients. In the future, it will be applied to other demographic groups.

Author’s response: Thank you for your valuable feedback. 

Reviewer 3

Comment 1: Since the paper involves a large number of abbreviations for simplicity, it is recommended that a list of such abbreviations be included.

Nutrient Adequacy Score (NAS)

Naturally nutrient-rich score 7 (NNR7)

Recommended dietary allowances (RDA)

being small for gestational age (SGA)

Food Composition Table of Bangladesh (FCTB)

Food-based dietary guidelines of Bangladesh (FBDG)

Linear programming (LP)

Estimated energy requirement (EER)

Physical activity level (PAL)

Tolerable Upper Limit (TUL)

Acceptable Macronutrient Distribution Range (AMDR)

Author’s response: Thank you for your comment. We have made a list of abbreviations before the introduction section in the revised manuscript. 

Comment 2: It is necessary that each equation be assigned a number.

Author’s response: Thank you for your comment. We have inserted numbers to the equations in the revised manuscript as can be seen in line numbers 149, 160, 176, 192, and 195.

Comment 3: Folic acid deficiency is typically regarded to be more common than calcium deficiency among pregnant women. Inadequate folic acid intake during pregnancy is connected with a number of potential health problems. What are these risks? Why the authors were folic acid excluded from the study, even though the need for folic acid is important at all stages of pregnancy? As folic acid is no less important than calcium, which was mentioned in this study.

Author’s response: Thank you for the comments. You are right. However, we used cut-off points in identifying the number of nutrients that are at risk of consumption (based on EAR and using probability approach) by the Bangladeshi population. The risk of dietary inadequacy of folic acid was found 67% for women of reproductive age and 50% for pregnant women (DOI: 10.13140/RG.2.2.31298.56002). We used cut-off points for the risk of dietary inadequacy above 80% which means that more than 30% of the population needed additional nutrient consumption to reduce the prevalence of risk of inadequacy back to 50% of that specific nutrient. That’s why folic acid was not considered a problem nutrient. 

Comment 4: In line 26 NNR7, 98, 108, 238, 246, 308, 318 NNR7, these terms new naturally nutrient-rich score, Naturally Nutrient Rich 7, new Nutrient-Rich Foods (NRF), and nutrient density scoring matrix are not new and you should specify with new metric for pregnant women in Bangladesh.

Actually, these terms had been used in various research studies and publications to assess the nutritional quality of different foods and dietary patterns. The United States Department of Agriculture (USDA) created a nutrient profiling technique known as the Nutrient-Rich Foods Index in order to evaluate the overall nutritional content of different foods. It assigns a ranking to each food item based on its nutrient density, considering both the good and negative aspects of the foods' nutrient composition when doing so.

Author’s response: Thank you for your comment. We have specified throughout the revised manuscript that the formulated scoring matrix has been developed for third-trimester pregnant women for the first time in Bangladesh.

Comment 5: In line 27, the phrase “third trimester pregnant women” should be added to the “specifically for pregnant women…”.

Author’s response: We have addressed this comment in the revised manuscript. Please see lines 13-15. 

Comment 6: In line 62, remove the short letters of the being small for gestational age (SGA) as it didn’t repeat it within the entire manuscript.

Author’s response: We have removed the abbreviation in the revised manuscript. 

Comment 7: In line 100, the authors only targeted the later stages of pregnancy; they did not target the entire gestation period. Therefore, only referring to pregnant women in their third trimester would be the most accurate course of action to pursue.

Author’s response: Thank you for your comment. We have added the third-trimester pregnant women in the mentioned line. Please see line 102-104. 

Comment 8: In line 115, delete “and” of the and egg.

Author’s response: Thank you for your comment. The term “meat, poultry, and egg” all together is a name of a food group. We have separated the names of the food groups with double-inverted commas for better understanding in the revised manuscript. Please see lines 116-117 and 119-122. 

Comment 9: In line 117-120, the sentence needs reference (such as FAO or FANTA,...etc).

Author’s response: Thank you for your comment. We have added a reference to the mentioned lines. Please see the lines 119-122. 

Comment 10: In line 134 table 1, the references should mention the recommended dietary allowance of pregnant women and which allowances were followed (icmr-NIN and/or others).

Author’s response: Thank you for your comment. We have addressed this comment in the revised manuscript. 

Comment 11: In line 146, the authors followed a specific food group classification. The classification of food groups can vary depending on the system or organization you should referring to! However, one common classification system is based on the United States Department of Agriculture (USDA) food groups, which are used for dietary guidelines and nutritional recommendations.

Author’s response: Thank you for your insight. We followed the food group classification of the food-based dietary guideline of Bangladesh. However, we modified the food group as described in lines 115-122 of the revised manuscript. 

Comment 12: In line 163, the word light should be change with sedentary.

Author’s response: We have changed the word light to sedentary. 

Comment 13: In line 172, the phrase ‘adverse effects on excess consumption’ is belong to the pregnant women take any supplements or medication or from the natural food? What grad of the adverse effect from natural food? I would suggest to remove these phrases or justified.

Author’s response: Thank you for your feedback. We have removed the phrase ‘adverse effects on excess consumption’ from the updated manuscript. Please see lines 189-191. 

Comment 14: In line 223, in table 4 authors should mention to the Indian ICMR-NIN as they considered as a RDA reference

Author’s response: Thank you for your comment. We have addressed this comment in the revised manuscript. 

Comment 15: In line 224 table 4, food groups. Thiamine is a vitamin that can be degraded when exposed to high temperatures since it is heat-sensitive. However, the amount of thiamine that actually breaks down during baking might be somewhat variable. Such as... * Note: Thiamine is sensitive to heat; therefore, grains and baked goods are not considered the primary sources of thiamine.

Author’s response: Thank you for highlighting the point. You are right. However, you know that the estimated average requirement (EAR) for thiamine was taken from updated guidelines (Nutrient requirements for Indians, NIN, 2020) which adjusted cooking losses of thiamine for establishing EAR and RDA for Indians.

Reviewer 4

Comment 1: The authors have claimed that the metric used to calculate the nutrient density of foods is completely new. NNR7 is not a completely new metric, it's an NNR version in which the nutrients selected are different from the original version. I suggest citing and explaining the original version of NNR (Drewnowski A. Concept of a nutritious food: Toward a nutrient density score. Am J Clin Nutr.368 2005;82(4):721–32) first (in the method section) and then explaining how and why this version (NNR7) was developed in detail.

Author’s response: Thank you for your feedback. We have addressed this in the method and introduction section in the revised manuscript. Please see lines 128-134 and 76-79. 

Comment 2: Please clarify how NNR has been calculated step by step. The following is the original formula developed by Drewnowski (Drewnowski A. Concept of a nutritious food: toward a nutrient density score. Am J Clin Nutr 2005;82(4):721-32.) NNR=Σ %DV2000 kcal /14 the formula presented here could not be correct. NNR is the mean of DV for selected nutrients, not RDA.

Author’s response: Thank you for your comment. In our NNR7 calculation, we have used the RDA values instead of DV because, for the Bangladeshi population, there are no standard DVs for nutrients. Thus, we had to borrow the RDA of the nutrients of the Indian population. The step-by-step calculation of NNR7 has been written in lines 138-149 of the revised manuscript. 

Comment 3: In this study, three nutrient-adequate diets have been formulated for third-trimester pregnant women. Although the method used to produce these diets is interesting, I think the formulated diets are not that applicable, even as an example. The foods and the amounts calculated by linear programming may not practically form a real daily menu. The alternative approach could be calculating the amount of serving size for each food group, then highlighting the food with the highest nutrient density in each food group to fulfill requirements for selected nutrients. If such results are achievable by linear programming, I recommend reporting them instead of what has been presented as a nutrient-adequate diet.

Author’s response: Thank you for expressing your concern regarding the applicability of the formulated diets. Please note that, to ensure cultural acceptability, the LP model was constrained to have rice, wheat, potato, and soybean oil in amounts that minimize deviation from the consumption pattern of pregnant women in Bangladesh. Also, the LP model was based on constraints that forced the formulated diets to have amounts of foods from various food groups that fall within the range recommended by the dietary guidelines in Bangladesh. The results of the formulated diets reflect that locally available nutrient-rich foods can be combined to meet the nutrient requirements and dietary diversity as recommended in the dietary guidelines. However, we agree that the formulated diets are not to be followed each day of the month or year. In line with the reviewer’s suggestion, we have thus also provided in Table 4 a ranked list of foods from each food group based on their nutrient density (i.e., level of contribution to meet the RDA of the seven problem micronutrients).

---

## [Decision Letter · Decision Letter 1]

20 Dec 2023

Nutrient Density of Bangladeshi foods and its application in diet planning for pregnant women

PONE-D-23-13289R1

Dear Dr. Shaheen

We’re pleased to inform you that your manuscript has been judged scientifically suitable for publication and will be formally accepted for publication once it meets all outstanding technical requirements.

Kind regards,

Dharmendra Kumar Meena

Academic Editor

PLOS ONE

Additional Editor Comments (optional):

The article can be accepted for publication

Reviewers' comments:

Reviewer's Responses to Questions

**Comments to the Author**

1. If the authors have adequately addressed your comments raised in a previous round of review and you feel that this manuscript is now acceptable for publication, you may indicate that here to bypass the “Comments to the Author” section, enter your conflict of interest statement in the “Confidential to Editor” section, and submit your "Accept" recommendation.

Reviewer #2: (No Response)

Reviewer #4: All comments have been addressed

2. Is the manuscript technically sound, and do the data support the conclusions?

Reviewer #2: Yes

Reviewer #4: (No Response)

3. Has the statistical analysis been performed appropriately and rigorously? 

Reviewer #2: I Don't Know

Reviewer #4: (No Response)

4. Have the authors made all data underlying the findings in their manuscript fully available?

Reviewer #2: Yes

Reviewer #4: (No Response)

5. Is the manuscript presented in an intelligible fashion and written in standard English?

Reviewer #2: Yes

Reviewer #4: (No Response)

6. Review Comments to the Author

Reviewer #2: Thank you for addressing the reviewer's comments. I have no additional comments regarding this manuscript.

Reviewer #4: (No Response)

7. PLOS authors have the option to publish the peer review history of their article (what does this mean?). If published, this will include your full peer review and any attached files.

Reviewer #2: **Yes: **AHMAD MOHAMED AMIN SALIM AL ATHAMNEH

Reviewer #4: **Yes: **Marziyeh Ashoori

---

## [Editor Report · Acceptance letter]

8 Jan 2024

PONE-D-23-13289R1 

PLOS ONE

Dear Dr. Shaheen, 

I'm pleased to inform you that your manuscript has been deemed suitable for publication in PLOS ONE. Congratulations! Your manuscript is now being handed over to our production team.

Kind regards, 

on behalf of

Dr. Dharmendra Kumar Meena 

Academic Editor

PLOS ONE